# Lactic Acid Bacteria in Dairy Foods: Prime Sources of Antimicrobial Compounds

Nooshzad Ahansaz [1], Armin Tarrah [2,*], Shadi Pakroo [1], Viviana Corich [1] and Alessio Giacomini [1]

[1] Department of Agronomy Food Natural Resources Animal and Environment (DAFNAE), University of Padova, Viale dell'Università, 16, 35020 Legnaro, PD, Italy; alessio.giacomini@unipd.it (A.G.)
[2] Canadian Research Institute for Food Safety, Department of Food Science, University of Guelph, Guelph, ON N1G 2W1, Canada
[*] Correspondence: atarrah@uoguelph.ca or tarrah.armin@gmail.com

**Abstract:** This review presents an in-depth examination of fermented dairy products, highlighting their significance as rich sources of antimicrobial agents. Through a comprehensive study of microbial activities during fermentation, we identify and discuss the rise of bioactive elements with antimicrobial characteristics. Bacteriocins such as nisin and pediocin play a significant role, as do organic acids such as lactic and acetic acid in providing antimicrobial activity. Challenges, including the enzymes, heat and pH sensitivity of certain compounds, are also touched upon, emphasizing the need for stable delivery for consistent efficacy. Our discussion covers various compounds, including bacteriocins, organic acids, and bioactive peptides, detailing their functions, action mechanisms, and potential applications. Moreover, this review discusses the emerging role of genetic engineering in optimizing lactic acid bacteria strains and exploring the potential of genetically modified organisms in improving the production and efficacy of antimicrobial compounds in dairy products. Additionally, we emphasize the interaction between beneficial microbes and their antimicrobial byproducts and discuss strategies for enhancing the synthesis of these antimicrobial compounds. The review highlights the nutritional significance of fermented dairy items and their potential as a rich source of compounds crucial for improving food safety. Additionally, the review explores challenges and potential solutions related to the stability of these compounds, ensuring their consistent efficacy and contribution to overall well-being.

**Keywords:** lactic acid bacteria; dairy foods; antimicrobial compounds; bacteriocins; safety

## 1. Introduction

The food and beverage sector can be categorized into the production and the subsequent distribution of these products [1]. To minimize losses at different stages, the food and beverage industry prioritizes two strategies: firstly, by introducing novel flavors to attract consumers and create demand; and, secondly, by improving the shelf life of products to reduce wastage and enhance overall efficiency [1]. Food fermentation is one of the most effective techniques for transforming fresh foods into various products [1]. Fermentation improves the content of essential amino acids, vitamins, flavor, and the aroma of food products. Additionally, fermented foods often require less cooking or heating compared to their non-fermented counterparts, leading to energy savings [2]. Studies have demonstrated that fermented foods possess superior quality and higher nutritional content compared to unfermented ones, primarily due to the presence of beneficial microorganisms [3]. Fermented foods offer numerous benefits, primarily due to the active involvement of microorganisms such as bacteria, yeasts, and molds. These organisms provide essential enzymes and metabolic activities that transform raw ingredients into digestible forms, improving nutrient absorption [4]. This fermentation process not only elevates enzyme content, which aids digestion and nutrient absorption [5], but also enhances the bioavailability of certain

nutrients, optimizing their absorption and utility in the body [6]. Beyond the health benefits, fermentation introduces new flavors that enrich the culinary experience. An example of microbial transformation is lactic acid fermentation, which is used to acidify milk and produce fermented dairy products such as yogurt, cheeses, and butter [7]. Lactic acid bacteria (LAB) are regarded as 'Generally Recognized as Safe' (GRAS) and are commonly used in the dairy industry and also form part of the microbiota of the human intestine [8]. LAB play a significant role in biopreservation because they produce a variety of antimicrobial metabolites during the development and fermentation processes [9]. The use of antimicrobial-producing LAB in the production of dairy products, which can be incorporated into fermented or nonfermented dairy products, implies a processing advantage to improve the safety and quality of dairy products, providing an additional barrier against foodborne diseases [10]. Among the most common antimicrobials are bacteriocins, which are ribosomally produced antimicrobial peptides. They can kill or inhibit undesirable bacterial strains, whether closely related or not, without harming themselves. This ability is especially relevant in the food industry. Notably, many LAB bacteriocins, including those derived from such bacteria, have shown efficacy against *Listeria monocytogenes*, a significant concern in traditional cheeses made from raw milk [11].

However, a study by Silva et al. [11] has shown that bacteriocins efficacy in food systems is frequently limited due to many problems, such as adsorption to food components and enzymatic degradation. For instance, in a study by Krishnamoorthi et al., nisin, a bacteriocin produced by *Lactococcus lactis* subsp. *lactis* strain CH3, was examined for its stability against various enzymes, including proteinase K and trypsin. As anticipated, given that bacteriocins are peptides, the bacteriocin lost its antibacterial activity upon treatment with these proteolytic enzymes [12] . The potential advantages of using bacteriocin-producing bacteria directly in dairy products could be an effective solution. Employing these bacteria can bypass the challenges associated with the inherent sensitivity of purified bacteriocins to proteolytic enzymes, as the continuous production of bacteriocins in the food matrix can counteract their degradation [12].

This review assesses the spectrum of antimicrobial compounds sourced from fermented dairy foods. Our exploration encompasses their variety, functionality, and cutting-edge techniques for enhanced production. Furthermore, we discuss challenges to integrate these compounds in dairy products alongside potential solutions to boost their stability and functionality. Our objective is to provide a comprehensive insight into these compounds, highlighting their important role in improving dairy product safety and quality while paving the way for novel research directions and applications within the dairy sector.

## 2. Bacteriocins in Dairy Foods

Bacteriocins are ribosomally synthesized short-length antimicrobial peptides produced by various groups of bacteria, especially LAB [13]. Bacteriocins produced by LAB are peptides mainly active against Gram-positive bacteria, including foodborne pathogens and food spoilage-related bacteria. Bacteriocins are categorized into various classes considering factors such as molecular size, physical properties, and the organisms that produce them. Class I comprises lantibiotics, due to the content of the unusual amino acid lanthionine, which are reputed for their heat stability and low molecular weight (around 5 kDa) [14]. Nisin A and its variants are the foremost examples of lantibiotics and have been the subject of extensive research [10]. Class II bacteriocins are distinguished by their simpler structures compared to lantibiotics. This class encompasses small, heat-stable peptides (around 10 kDa) that exhibit an amphiphilic helical conformation. They are further subdivided into three subclasses: II-A, II-B, and II-C [15]. Subclass II-A members stand out due to their potent antibacterial properties. These bacteriocins typically consist of 37–48 amino acid residues. Examples of this subclass include pediocins and enterocins [15]. Bacteriocins of subclass II-B, known as heterodimeric bacteriocins, are composed of two peptides. Lactococcin was the inaugural bacteriocin identified within this group. The mode of action involves dissipating the membrane potential and causing a reduction in the intracellular

concentration of ATP [15,16]. Subclass II-C bacteriocins are characterized by their circular configuration, arising from the covalent bond between the C and N terminals. This leads to peptides adopting a cyclic tail conformation. Their action mechanism involves permeabilizing the cytoplasmic membrane of target cells, ultimately leading to cell lysis [15]. Bacteriocins in Class III are large, thermolabile proteins with molecular weights exceeding 30 kDa. A defining feature of this class is their ability to induce cell wall lysis in target microbes [15]. Colicin, produced by *Escherichia coli*, serves as a representative example of Class III bacteriocins [15]. Furthermore, helveticin M is produced by *Lactobacillus crispatus*, helveticin J by *Lactobacillus helveticus*, and enterolysin A by *Enterococcus faecalis*, all falling under the category of Class III bacteriocins [17–19]. Finally, Class IV bacteriocins are characterized by their composition, which includes complex proteins conjugated with lipids or carbohydrates. Examples from this class include pediocin N5p and lactocin 27 [20].

The diverse classification of bacteriocins, ranging from heat-stable lantibiotics to complex protein conjugates, underscores their potential versatility in applications, particularly in dairy food preservation. These peptides, especially those produced by LAB, have received attention by the dairy industry for their capacity to combat foodborne pathogens and spoilage-related bacteria. A prime example is nisin, a bacteriocin produced by *Lactococcus* and some *Streptococcus* strains, renowned for its antimicrobial activities [21]. Additionally, LAB from *Lactobacillus* and *Leuconostoc* have been identified as prominent producers of class II bacteriocins, expanding the spectrum of potential antimicrobial agents [22]. Particularly noteworthy among these is the pediocin produced by *Pediococcus*, classified under class IIa bacteriocins, pediocin exhibits pronounced anti-listerial activity, finding substantial effectiveness in meat products [23]. In fact, there is an impressive array of approximately 30 class IIa bacteriocins that have been recognized, sourced from a variety of LAB genera including *Bifidobacterium*, *Lactobacillus*, *Lactococcus*, *Pediococcus*, *Leuconostoc*, *Streptococcus*, and *Enterococcus* [22].The *Enterococcus* species, for instance, produce enterocins that are particularly effective against strains like *Bacillus* and *Clostridium* species [24,25]. Studies also indicated that enterocin-producing *Enterococcus faecalis* strains exhibit inhibitory effects against *L. monocytogenes*, a pathogen of concern in fresh cheese [25].

In this light, three strategies can be used to consider LAB and bacteriocins for natural preservation in the food sector. Each of these strategies offers distinct applications and implications within the industry. Firstly, incorporating LAB for dairy fermentation and protection: incorporating LAB directly into dairy products exploits the benefits of natural fermentation, enhancing flavor, texture, and nutritional value [26]. As these bacteria grow and proliferate within the dairy matrix, they consistently produce bacteriocins, ensuring a sustained defense against spoilage microorganisms and pathogens. Furthermore, the use of LAB resonates with modern consumers who are increasingly seeking cleaner labels and natural preservation methods. Nonetheless, the effectiveness of this biopreservation approach is intricately tied to the viability and activity of LAB. Factors such as pH, temperature, and the competitive dairy microbial environment can influence their performance [27]. Moreover, the use of LAB in food products can greatly influence flavor profiles. Specific strains of LAB, such as *Streptococcus thermophilus* and *L. lactis*, are approved and commonly used to enhance flavors in dairy products. These strains are known to introduce a tangy and pleasant acidic taste, often associated with fermented dairy products such as yogurt. On the other hand, certain strains, such as *Limosilactobacillus fermentum* and *Limosilactobacillus reuterin*, if not used judiciously, can produce flavors that may be perceived as off or less palatable. This can be attributed to the production of certain metabolites such as biogenic amines (e.g., histamine), diacetyl (which in excess can impart a strong buttery flavor), or acetic acid (which can give a sharp, vinegar-like taste [28]). Secondly, there is the direct application of pure bacteriocin to dairy products. Introducing purified bacteriocins into dairy products ensures a uniform and immediate antimicrobial defense. This method eliminates the waiting period associated with bacterial growth, providing prompt protection against potential spoilage agents and pathogens. Moreover, since it is the isolated compound being added, there is minimal risk of introducing unintended flavors,

maintaining the original taste profile of the dairy product. However, there are challenges to this approach. The extraction and purification processes for bacteriocins can drive up production costs. Additionally, while some bacteriocins possess a broad antimicrobial spectrum, others might be narrowly effective, requiring precise targeting of threats. There is also the concern of potential degradation due to the presence of proteolytic enzymes in certain dairy products, which might compromise bacteriocin structure and efficacy [12]. Lastly, using bacteriocin-producing fermented products in dairy: using fermented products derived from bacteriocin-producing strains offers a unique approach to dairy preservation. Incorporating such products can provide dual benefits, such as introducing rich flavors from fermentation and conferring the antimicrobial properties of the bacteriocins present. This approach differs from the first one because the initial method directly adds LAB to the dairy products, initiating immediate fermentation and bacteriocin production. In contrast, the latter incorporates already fermented products, capitalizing on existing bacteriocins without initiating a new fermentation process in the dairy product itself. This method also promotes sustainability, utilizing byproducts or excess from one dairy process to enhance another. However, this approach has challenges. The concentration and activity of bacteriocins in these fermented products can vary, potentially leading to inconsistent preservation outcomes. Depending on the fermented source, strong flavors might also be introduced that could overshadow the desired taste profile of the final dairy product [29].

Expanding on this concept, there are clear examples of bacteriocins effectively used in dairy products to improve their quality and safety. For instance, lactacin F from *Lactoplantibacillus plantarum* has been employed in yogurt to improve its safety [30]. In a model fresh cheese, lacticin 481 produced by *L. lactis* L3A21M1 effectively reduced *L. monocytogenes* presence [31]. Skim milk utilized nisin Z from *L. lactis* W8 to extend its shelf life [32]. Additionally, nisin A was added to cottage cheese to inhibit the growth of *L. monocytogenes* [33]. Nisin, produced by *L. lactis* subsp. *lactis*, is a noteworthy bacteriocin among those derived from LAB. Recognized for its broad-spectrum activity against Gram-positive bacteria and its proven safety for human consumption, it has been approved as a natural food preservative in numerous countries. Dairy products, especially milk, are among its main applications [34]. Nisin is structured with unique amino acids, which give rise to its distinctive rings formed by thioether bonds [34]. Incorporating nisin into dairy products is an effective strategy to inhibit the proliferation of notable pathogens, including *L. monocytogenes* and *Staphylococcus aureus* [35]. Nisin exerts its antibacterial properties through a dual-action mechanism (Figure 1). Firstly, it binds to the cell wall precursor, lipid II, inhibiting cell wall synthesis. Simultaneously, this binding facilitates the formation of pores in the bacterial cell membrane, disrupting the integrity and leading to cell death [27,36].

Studies have shown that bacteriocins, such as leucocin A and sakacin A, are frequently utilized to counteract *L. monocytogenes* in dairy items [37,38]. Meanwhile, nisin, produced by *L. lactis*, has gained approval as a food preservative in several nations to suppress harmful bacteria in foods like ricotta cheese and other processed cheeses [39]. A study introduced a novel class III bacteriocin, NX371 from *L. acidophilus* NX2-6. This bacteriocin presents promising for its use in preserving milk and Mozzarella cheese, as it notably curtails pathogenic proliferation in dairy items. Furthermore, its resistance to heat and varying pH levels surpasses that of nisin [40]

Beyond specific applications of bacteriocins in dairy, these compounds have broader implications for food safety. The overall advantages of bacteriocins in food preservation underscore their significance. They can be degraded by proteases in the digestive tract, reducing consumer concerns about accumulation [41]. Moreover, they demonstrate effectiveness against a range of spoilage and pathogenic bacteria. In a study by Ageni et al. [42], bacteriocins displayed robust antagonistic effects against several pathogens, with *Pseudomonas aeruginosa*, *E. coli*, and *Clostridium* spp. being particularly sensitive. Thermal stability is another important characteristic of bacteriocins, aligning well with food processing requirements. Many food processing methods involve heat, making the heat stability of bacteriocins a valuable trait for preservation. Confirming this, both Moigani and

Amirinia, and Ageni et al. noted the remarkable ability of bacteriocins to maintain their antimicrobial or preservative function for 15 minutes at 121 °C, an attribute that proves invaluable in food safety procedures based on heat treatments [42,43]. The comprehensive study by Ibarra-Sánchez et al. [35] sheds light on both the challenges and emerging practices in utilizing nisin for dairy product conservation. The study confirms the efficacy of nisin in dairy products through various strategies, including antimicrobial packaging, bioengineering, encapsulation, and combined antimicrobials. Specifically for cheese, active antimicrobial packaging with nisin offers a robust defense against contamination either during or after processing. However, it must be noted that its protective action is confined to the cheese's surface. Addressing this limitation, incorporating encapsulated nisin into antimicrobial packaging emerges as a promising technique. This approach could elevate both stability and microbiological safety, ensuring the preservation of food's nutritional and sensory quality [35].

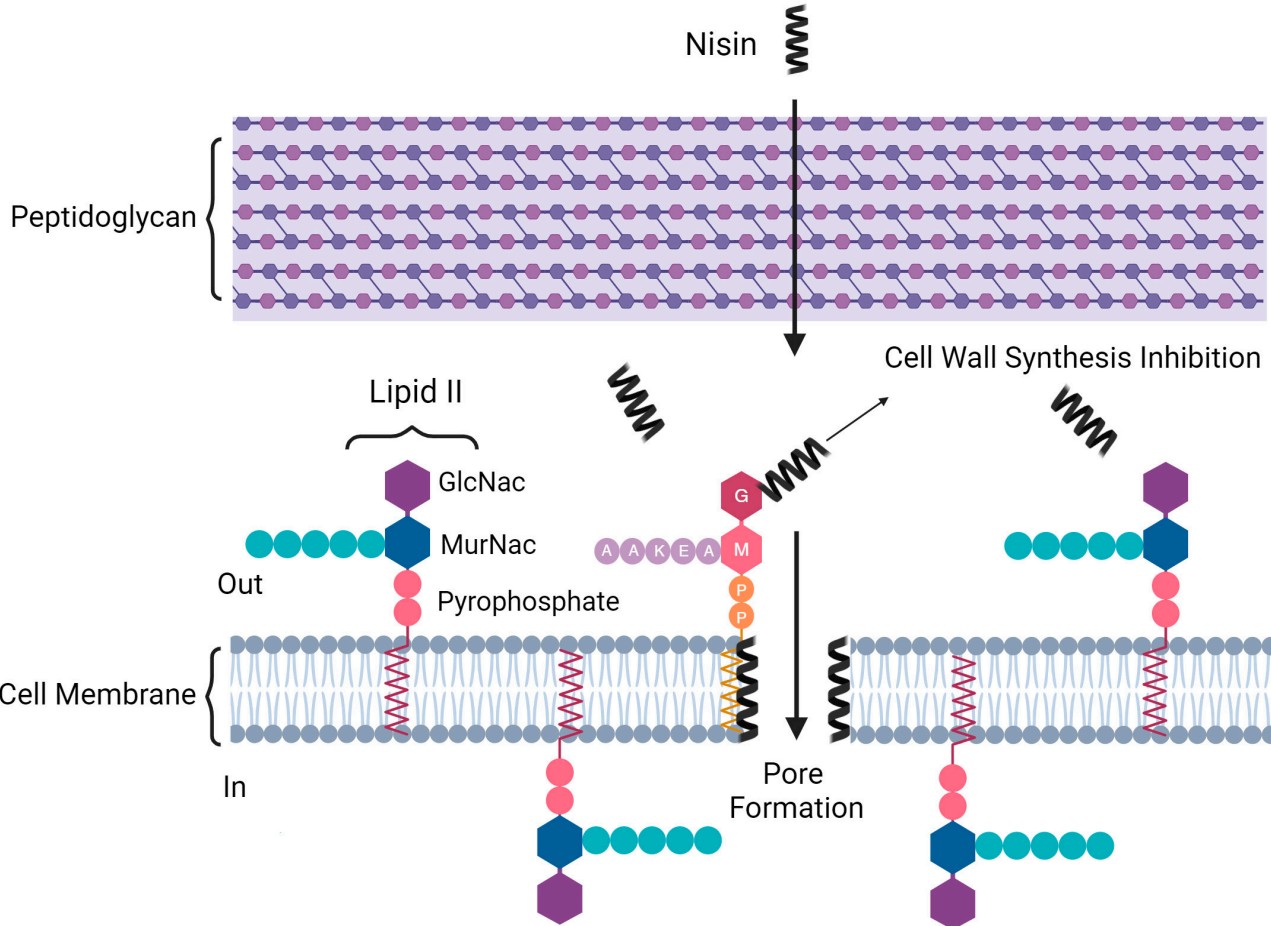

**Figure 1.** Mechanism of action of nisin [27,36].

In conclusion, bacteriocins present a promising alternative for natural preservation of dairy products. Their diverse classification and modes of action make them suitable for various applications within the dairy industry. While challenges remain in their application, their effectiveness against a broad spectrum of pathogens, combined with consumer preference for natural preservatives, places them at the forefront of dairy research and application. As food safety remains a primary concern, the strategic use of bacteriocins can significantly enhance the quality and longevity of dairy products. Furthermore, it is crucial to explore the untapped potential of relatively uncharted LAB strains in dairy foods. These lesser-studied strains hold unique characteristics and have the capacity to revolutionize dairy product preservation. Investigating these strains introduces a novel perspective to

the field, shedding light on their distinctive antimicrobial properties and their ability to contribute to innovative dairy products with unprecedented flavors and enhanced safety.

## 3. Organic Acids and Their Antimicrobial Properties

In fermented milk, lactic acid stands out as the primary organic acid, as result of the metabolic activity of lactose break down by LAB. However, depending on the specific LAB strains and their fermentation pathways, other organic acids, including formic and acetic acid, may also be present, especially from the processes of hetero-fermentative LAB [26]. LAB can operate as either homo-fermenters or hetero-fermenters. Homo-fermenters predominantly metabolize hexoses to produce lactic acid as the sole or primary fermentation end-product. Their metabolic pathway of choice is the Embden–Meyerhof–Parnas (EMP), often referred to as glycolysis. The EMP pathway is a linear process converting glucose to pyruvate, which is then reduced to lactic acid. This results in an efficient lactic acid production, creating an acidic environment that not only helps in food preservation but also gives the characteristic tangy taste to fermented foods [35]. Prominent examples of homo-fermenting LAB species include *Lactobacillus acidophilus*, *Lactobacillus bulgaricus*, *Ligilactobacillus salivarius*, *Lactobacillus helveticus*, *L. lactis*, and *S. thermophilus* [36]. In contrast, hetero-fermenting LAB have a more diverse metabolic output. They can produce lactic acid, acetic acid, alcohol, and carbon dioxide from their metabolic activities. These bacteria metabolize sugars through the phosphoketolase pathway. When breaking down pentose sugars, they typically produce lactic acid, acetic acid, and $CO_2$. For hexoses, the end products can be lactic acid, ethanol, and $CO_2$, with possible minor production of acetic acid. The carbon dioxide produced by hetero-fermenters is crucial in some food applications, such as leavening in sourdough bread. Examples of hetero-fermenting LAB include *Lentilactobacillus buchneri*, *Levilactobacillus brevis*, and *L. fermentum* [44,45].On the other side, facultative hetero-fermenters such as *L. plantarum*, exhibit remarkable metabolic versatility. When metabolizing hexose sugars such as glucose, they can utilize the EMP pathway for homo-fermentation, producing primarily lactic acid, or switch to the phosphoketolase pathway for hetero-fermentation, leading to lactic acid, ethanol, and $CO_2$. The choice between these pathways is influenced by various factors: environmental conditions such as pH and oxygen levels; substrate concentrations, which can shift the balance of end-products; and the strain's genetic makeup that predisposes it to prefer one pathway over another. In slightly acidic conditions, they might favor homo-fermentation, as it produces lactic acid, further inhibiting competitors. In more neutral conditions, hetero-fermentation can be advantageous, producing varied metabolites such as ethanol and $CO_2$, which offer different ecological benefits. Oxygen presence can also affect the choice. Limited oxygen might push LAB towards homo-fermentation for efficient energy production, while trace oxygen levels might tilt the balance towards hetero-fermentation to maintain cellular redox balance through ethanol production. Moreover, high glucose concentrations often steer LAB towards homo-fermentation, maximizing ATP production. However, a mix of sugar types, such as hexoses and pentoses, can push them towards hetero-fermentation, exploiting the diverse sugar array. Furthermore, the inherent genetic predisposition of the strain plays a definitive role. Some LAB strains, due to specific genes or regulatory elements, might inherently lean more towards one pathway, regardless of environmental conditions. Finally, in the presence of pentoses such as xylose, these LAB typically employ the phosphoketolase pathway for hetero-fermentation, producing lactic acid, acetic acid, and $CO_2$. This metabolic flexibility provides an adaptive advantage, allowing them to efficiently process substrates in varied fermentation contexts, influencing both preservation and flavor nuances of the resulting products [44,45].

While the intricacies of the metabolic pathways employed by LAB highlight the vast spectrum of organic acids produced, it is essential to emphasize the direct implications these acids have on food aroma, safety, and preservation. Among these organic acids, acetic acid is a potent compound with notable antimicrobial properties. Not only does it play a role in imparting a distinct tangy flavor to fermented foods, but its efficacy against a range of

spoilage and pathogens, especially fungi, is commendable. The antifungal activity of acetic acid is vital in the context of dairy fermentations, where mold and yeast contaminations can compromise product quality and safety. Acetic acid exhibits antifungal properties in dairy products primarily by penetrating the fungal cell membrane in its undissociated form. Once inside the cell, it dissociates into acetate ions and protons, leading to a decrease in intracellular pH. This acidic environment disrupts essential enzymatic reactions, hindering the fungal cell metabolic processes. Additionally, the osmotic imbalance caused by the accumulation of acetate ions compromises the integrity of the cell membrane, resulting in leakage of cellular contents and inhibiting fungal growth [46]. Beyond acetic acid, there's Phenyllactic Acid (PLA), which is synthesized from the amino acid phenylalanine and demonstrates broad antifungal activity by disrupting fungal cell membranes and energy metabolism. Furthermore, organic acids such as lactic acid, formic acid, and propionic acid, each contribute uniquely to the antifungal effect. For instance, formic and propionic acids penetrate fungal cells and disrupt their internal pH balance and metabolic functions. Recognizing the diverse antifungal arsenal of LAB, encompassing acetic acid, PLA, and other organic acids, is crucial. Their collective and potentially synergistic effects can significantly enhance the preservation and safety of dairy products [46]. Delving deeper into the antimicrobial mechanisms of acetic acid and other organic acids can provide insight into their potential applications in enhancing the shelf life and safety of dairy products. Fungal contamination restricts the shelf-life of fermented dairy products, leading to both food wastage and economic challenges [46]. While conventional methods such as heat treatment and air filtration are employed to decrease contamination during the production [47], there is an increasing consumer preference for natural preservation methods, such as biopreservation [48]. In this context, LAB strains exhibiting antifungal properties present a promising solution. Research indicates that certain LAB strains, including *L. plantarum*, *Lacticaseibacillus rhamnosus*, and *Lacticaseibacillus casei*, possess inherent antifungal activities [48–51]. The primary antifungal mechanisms attributed to LAB in dairy settings are the synthesis of specific antifungal compounds and a reduction in pH [46]. Of these compounds, acetic acid, a metabolite produced by LAB, stands out for its significant antifungal effects in dairy products [51]. Intriguingly, the inhibitory concentration of acetic acid is enhanced when in the presence of lactic acid, suggesting a synergistic action between the two [52].

Lactic acid, a primary byproduct of LAB metabolism, is instrumental in lowering the pH of fermented dairy products. This reduction in pH plays a critical role, resulting in a higher proportion of undissociated acetic acid and enhancing its antimicrobial potency [53]. Furthermore, Garnier et al. [54] conducted an in vitro research on three fermented dairy products produced (a reconstituted 10% low heat milk supplemented with 45% anhydrous milk fat and an ultrafiltration permeate supplemented with 1% yeast extract) by *Acidipropionibacterium jensenii* CIRM-BIA1774, *L. rhamnosus* CIRM-BIA1952, and *Mucor lanceolatus* UBOCC-A-109193. All three products demonstrated strong antifungal properties. Their investigation revealed propionic and acetic acid as the primary antifungal constituents, enhanced by lactic and butyric acids. Garrote, Abraham, and De Antoni [55] reinforced this observation, pinpointing acetic and lactic acids as the primary agents behind the antimicrobial strength of kefir. Further research by Lind, Jonsson, and Schnürer identified acetic and propionic acids as chief antagonists against fungi such as *Aspergillus fumigatus* and *Aspergillus nidulans*, whereas lactic acid exhibited a lesser effect. It is also important to note that the inhibitory potential of these acids on fungal growth tends to diminish with increasing pH [56].

Understanding antimicrobial implications of organic acids in dairy products is crucial, but so is the appreciation of their impact on the sensory and textural qualities of these products. Organic acid concentrations, particularly lactic acid, have far-reaching effects beyond inhibitory action against spoilage organisms. For instance, over-acidification in cheese is considered a defect, leading to an undesired crumbly texture and overly tangy flavor. Ensuring the right level of acid production is crucial not just for achieving the desired taste and texture but also for maintaining the structural integrity of the cheese [57–59].

Alongside understanding the critical balance of acid production in dairy, advancements in production techniques are equally important. Cubas-Cano et al. [60] further emphasized the possibilities brought forth by innovative process configurations. Techniques such as fed-batch and simultaneous saccharification and fermentation (SSF) not only streamline production but also enhance resource utilization efficiency, thereby potentially reducing the production costs. Fed-batch processing, a method where substrates are added incrementally to the fermentation vessel instead of a singular addition at the start, has become a key approach in modern fermentation processes. This strategy effectively manages microbial growth rates by adjusting substrate availability, avoiding problems such as overpopulation or running out of substrate. Additionally, adding feed over time can lengthen the production phase, possibly increasing yield. Another important method is SSF, where the breaking down of complex sugars and fermentation happen at the same time. This combined process removes the need for separate steps in breaking down sugars, making it more direct to turn complex sugars into end-products. Plus, as fermentation and the breakdown of sugars occur together, any potential harmful compounds produced are quickly used up or changed by the fermenting microbes [60]. The continuous evolution of these methods opens the door for more sustainable and efficient production pipelines in the food and biotech sectors. Beyond yield improvement, these advancements reflect the industry's broader goal: achieving high-quality, consistent, and scalable production that can cater to global demands while minimizing environmental footprints.

In dairy fermentation, the profound roles of organic acids are undeniable. Their dual function in enhancing preservation and flavor profiles is essential. As we explore the intricacies of dairy processing, balancing safety with sensory appeal is crucial. With current industry advancements, we stand at the threshold of a new era where we can sustainably, efficiently, and flavorfully meet global demands. In addition, our exploration should go beyond individual organic acids to understand how different organic acids synergize, potentially enhancing their antimicrobial effects on foodborne pathogens. This distinctive approach addresses the complexity of micro-bial interactions and opens new horizons for more effective strategies in preserving dairy products, thus adding an unprecedented layer of significance to this section.

## 4. Antimicrobial Compounds Diversity in Dairy Foods

Dairy fermented foods are common in many diets around the world. They are valued for their taste and texture and the many health-boosting compounds they contain. LAB particularly add these compounds during fermentation, making the food both healthy and safe. LAB are effective inhibitors of pathogens and significantly hinder the activities of food-spoiling organisms. When introduced to food, they can suppress harmful gut pathogens and reduce toxic elements in the intestines [61]. Additionally, LAB enhance the nutritional value and texture of dairy items, including yogurt and cheese. They achieve this while also fostering gut health through the production of antibacterial agents [62,63]. Commercially, LAB are popularly utilized as starter cultures due to their range of metabolic abilities. Besides the well-known bacteriocins and organic acids, LAB produce many other bioactive molecules that deserve attention in the context of dairy fermentation (Figure 2) [64,65]. Specifically, milk fermentation involves various LAB species such as *Streptococcus*, *Leuconostoc*, *Pediococcus*, *Lactococcus*, and *Lactobacillus* [63]. For instance, *L. rhamnosus* is effectively employed as a probiotic in fermented dairy products and its incorporation in cheese production could mitigate the risk of pathogen proliferation, speed up cheese maturation, and enhance cheese flavor [66].

Reuterin (3-hydroxypropionaldehyde) is one of the prominent examples, which is produced by some strains of *L. reuteri* during the fermentation of dairy products such as cheese and yogurt, and it exhibits broad activity against foodborne pathogens [67,68]. Beyond reuterin, compounds such as diacetyl also emerge during fermentation, adding to the diverse antimicrobial arsenal of LAB [67]. Reuterin works by inducing oxidative stress inside microbial cells. This compound has a multi-targeted mechanism, interacting

with thiol groups in enzymes and other cellular proteins, leading to impaired function. Additionally, reuterin disrupts microbial cell membranes, alters intracellular pH, and causes DNA damage [69]. On the other hand, diacetyl, a volatile compound naturally found in fermented dairy products, primarily exhibits its antimicrobial activity by lowering the internal pH of microbes. The acidic environment created by diacetyl destabilizes the microbial cell membrane, leading to a loss of essential molecules and eventual cell death [70]. The synergistic action of reuterin and diacetyl has been emphasized, with these natural antibacterials enhancing the safety of acidified dairy products [67]. Recent studies have also expanded our understanding of these compounds. Sun et al. [71] reported that the reuterin system regulates intestinal flora and has anti-infection, anti-inflammatory, and anti-cancer properties. Additionally, Ortiz-Rivera et al. [72] evaluated the antimicrobial potential of reuterin, both in vitro and as part of a fermented milk product. Their findings indicated a stronger susceptibility of Gram-negative bacteria compared to Gram-positive bacteria to reuterin's action. Significantly, the presence of reuterin did not compromise the quality aspects of fermented milk, such as pH and acidity, indicating its efficacy as a preservative agent. In a separate study, Assari et al. [73] investigated LAB isolates for their antibacterial effects against pathogens such as *L. monocytogenes* and *S. aureus*. Their results indicate significant antimicrobial activity from diacetyl production. Besides its antimicrobial properties, diacetyl also plays a role in hydrolyzing milk proteins, enhancing food digestibility, and contributing to flavor [74].

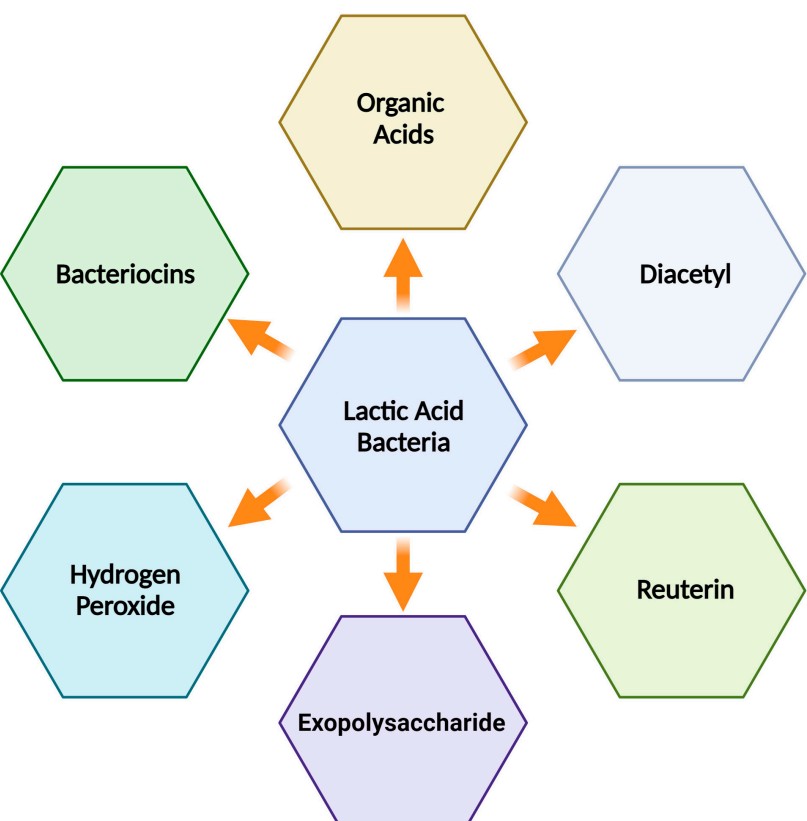

**Figure 2.** Diversity of antimicrobial compounds produced by LAB in dairy fermentation.

Furthermore, the lactoperoxidase system (LPOS) has been highlighted as a promising preservative agent to combat foodborne pathogens [75]. The lactoperoxidase system is a naturally occurring antimicrobial in milk. At its core is the enzyme lactoperoxidase, which catalyzes the oxidation of thiocyanate (SCN-) in the presence of hydrogen peroxide ($H_2O_2$), yielding hypothiocyanite ions (OSCN-). These ions are particularly reactive and play a crucial role in the antimicrobial action. The hypothiocyanite ions disrupt bacterial cell membranes by oxidizing the sulfur-containing amino acid residues in proteins and enzymes.

This oxidation damages bacterial membrane integrity, causing increased permeability, loss of vital molecules, and eventually cell lysis [76]. Al-Baarri et al. [75] examined the efficacy of the LPOS against *E. coli* in fresh cow milk and its related products to assess its antibacterial potency. They used enzymatic reactions, such as that with hydrogen peroxide, to derive the antimicrobial agents from LPOS. Their findings indicate that LPOS serves as a potent antibacterial agent.

The production of EPS by LAB has also gained attention in the food sector, specifically in dairy. These EPS influence the texture and consistency of fermented foods and play a vital role in providing antimicrobial benefits in such products [77]. Moreover, EPS represent a protective coating that can shield LAB by forming a physical barrier, blocking harmful bacteria, and trapping essential nutrients, giving EPS-producing bacteria an advantage over others [78]. Research by Angelin and Kavitha [79] and others has shown that EPS produced by LAB can exhibit protective effects both in vitro against a spectrum of pathogens, including those residing in the digestive system. However, the challenges of integrating EPS at industrial level include their modest yields and variability across different EPS-producing bacterial strains. Thus, there is a pressing need to explore new strategies for enhancing EPS production for prospective industrial uses [77]. While the potential of LPOS and EPS produced by LAB in dairy products has been increasingly recognized, several aspects require deeper exploration. The scalability of the use of LPOS as an antimicrobial agent in industrial dairy production is yet to be thoroughly evaluated. Given the global demand for dairy products and the constant need for safer preservation techniques, integrating natural antimicrobial agents such as LPOS could revolutionize dairy processing. Similarly, while the antimicrobial and textural benefits of EPS are evident, the exact mechanisms by which these compounds exert their effects are still being unraveled. This knowledge is crucial for manipulating LAB strains to consistently produce optimal EPS amounts and quality for specific dairy applications. The economic aspects of integrating these compounds on a large scale, such as cost–benefit analyses comparing traditional preservatives versus LPOS or EPS, are also essential to consider.

Beyond the direct production of antimicrobial agents such as EPS, LAB also employ sophisticated communication systems to coordinate their activities. This communication, known as quorum sensing (QS), represents another dimension of how these beneficial microbes influence the fermentation process and, in turn, the safety and quality of dairy products. Traits regulated by QS, including bacteriocin production and acid stress tolerance, influence not just foodborne pathogens but also the shelf-life of food products [80,81]. Central to QS systems is the synthesis of autoinducers (AIs), low molecular weight signaling molecules, which are recognized and responded to by nearby bacterial cells [82,83]. Notably, several bacteriocins, potent antimicrobial peptides produced predominantly by Gram-positive bacteria, are synthesized in a QS-regulated manner [84]. QS is well-recognized in food microbiology due to its association with foodborne pathogenicity, spoilage, and biofilm formation [85]. Numerous studies indicate that various QS mechanisms are present in fermented foods. This suggests that modifying the involved QS systems can positively influence the quality of fermented foods [80]. One study evaluated the AI-2 activity in lactic-fermented foods and found varying AI-2 signaling intensities. Based on these findings, it is understood that interactions that take place both within and between species in kimchi likely involve AI-2 signaling activities, potentially influencing product variety [85]. Additionally, AI-2 signaling in LAB enhances the beneficial properties of fermented foods. QS molecules identified in spoiled products influence microbial diversity and metabolic actions. These molecules could serve as essential markers for monitoring dairy product quality during storage and for preventing spoilage [86].

This suggests potential avenues to positively influence fermented food quality by modulating inherent QS systems [80]. A comprehensive understanding of QS in LAB reveals a tri-component system: an autoinducing peptide (AIP), a membrane-bound histidine kinase (HK) sensor, and an intracellular response regulator (RR) [87,88].

In summary, dairy-fermented foods harbor a rich assortment of antimicrobial compounds, significantly enhancing the nutritional, safety, and sensory attributes of the product. LAB play an instrumental role in producing these compounds, with reuterin, $H_2O_2$, diacetyl, LPOS, and EPS as just a few noteworthy examples. Their combined effects safeguard the products against harmful pathogens and impart unique textures and flavors that consumers appreciate. However, challenges remain, particularly in scaling up the use of these compounds for industrial applications and ensuring consistent production. Additionally, the complex QS mechanisms employed by LAB underscore the complexity and sophistication of microbial interactions during fermentation. Applying these interactions and antimicrobial agents offers promising opportunities for the future of dairy processing. Moreover, the exploration of microbial compounds should not be limited to well-known metabolites. We should also delve into underexplored LAB strains and microbial consortia. Traditional LAB strains remain integral, but in the context of the rapidly evolving dairy product landscape, in-depth exploration of these lesser-studied microbes and their metabolites becomes imperative. This pioneering approach uncovers innovative possibilities, creating dairy products with distinctive textures, flavors, and functional attributes that resonate with contemporary consumer expectations. As the dairy industry evolves, interdisciplinary collaboration will be essential in ensuring a seamless and effective transition from laboratory findings to industrial application.

## 5. Safety Assessment of LAB and Bioactive Compounds in Dairy Foods

LAB have always played a central role in dairy product fermentation, mainly due to their capacity to produce advantageous compounds such as bacteriocins and organic acids [89]. However, their longstanding association with food processing and potential health benefits does not automatically confer a universally accepted safety status to all LAB species and strains. A rigorous evaluation of their safety profile becomes crucial when we investigate the potential of LAB for dairy and other food products. Central to this safety assessment in the US is the GRAS designation, for which the US Food and Drug Administration (FDA) is the primary authority (https://www.fda.gov/food/food-ingredients-packaging/generally-recognized-safe-gras; accessed on 7 June 2023). It is the FDA's responsibility to scrutinize and grant the GRAS status, which underscores the necessity for a comprehensive examination of each LAB strain safety profile [90–92]. Meanwhile, the European Food Safety Authority (EFSA) uses the Qualified Presumption of Safety (QPS) status for equivalent evaluations in Europe [93,94]. This method defines a standardized safety benchmark for microorganisms intended for human consumption.

But beyond the FDA GRAS and EFSA QPS designations, certain strain-dependent traits of food-grade microbes, such as the presence of acquired antibiotic resistance, strain-dependent virulence features, and genome stability, in any case need a meticulous evaluation before their deliberate incorporation into food systems [95] (Figure 3).

While regulatory frameworks offer a broad guideline, practical evaluations have revealed a detailed understanding of LAB strains in real-world applications. Recent studies have shed light on the safety evaluation of LAB strains, adding depth to the theoretical framework. For instance, Kim et al. [96] focused on the safety evaluation of *L. lactis* IDCC 2301, a strain isolated from homemade cheese. Utilizing both in vitro and in vivo assays, they evaluated aspects such as antibiotic resistance and potential toxin production. The results were conclusive, revealing that *L. lactis* IDCC 2301 was devoid of toxigenic genes and exhibited no antibiotic resistance. Thus, it was deemed truly safe for its utilization as probiotic for human consumption. In another study, Colombo et al. [92] conducted a comprehensive study examining the safety features of 15 LAB strains, all of which had been previously isolated from a dairy environment. The research delved into the production of potential virulence factors and assessed resistance against relevant antibiotics. Their findings were reassuring since the selected strains appeared relatively safe for integration as beneficial cultures in the food industry. None of the strains revealed phenotypical production of any tested virulence traits or of biogenic amines. Interestingly, antibiotic

resistance was noted for gentamicin, clindamycin, vancomycin, rifampicin, erythromycin, tetracycline, and ampicillin. However, the methods employed in that study do not differentiate between acquired and intrinsic resistance. While intrinsic resistance is not deemed detrimental and can be even advantageous for the beneficial strain, acquired resistance presents a risk as it can be transferred to other microbes, potentially leading to public health concerns. The Study by Tarrah et al. [97] identified an acquired resistance gene for tetracycline in *Streptococcus thermophilus*, a well-recognized microorganism employed in yogurt production. While the strain indicated desirable traits, such as a high amount of folic acid production, the presence of this acquired resistance gene stopped further investigation. A genomic analysis of this strain revealed a notable similarity of this gene with a gene with the same function in *L. monocytogenes*. Moreover, an identified phage region on DNA provided evidence of past transmission events. This discovery underscores the vital importance of strain-specific safety evaluations, highlighting that even well-characterized and widely used strains can harbor unexpected genetic elements with potential implications for safety. Transitioning from the issue of acquired resistance, Tarrah et al. [98] undertook an insightful study on four *Streptococcus gallolyticus* subsp. *macedonicus* (*S. macedonicus*) strains isolated from dairy environments. Their in-depth genomic analysis revealed several genes associated with virulence activity and evidence of unstable genomes potentially arising from horizontal gene transfer. This latter finding underscores the complexity and dynamism of microbial genetics, especially in the context of potential food applications. Notwithstanding these safety concerns, the strains of this species have attracted scientific interest due to their notable technological attributes, such as specific enzymatic activities. Yet, despite their potential, strains from *S. macedonicus* have not achieved GRAS or QPS status to date. Research continues to further elucidate the balance between their promising technological features and safety aspects.

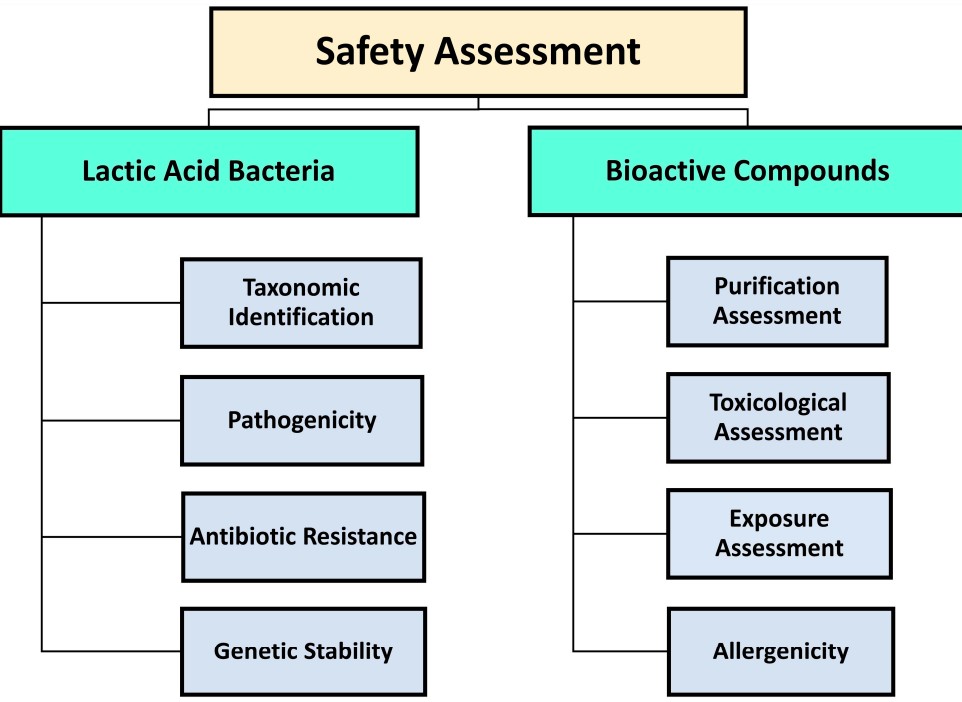

**Figure 3.** Assessment procedure for LAB and bioactive compounds in dairy fermentation.

Yet, as science delves deeper into the potential applications of LAB, it is not just the bacteria to be under meticulous evaluation but also the bioactive compounds they produce. It is crucial to analyze safety aspects, including toxicological properties, allergenicity, purity, and potential exposure risks of bioactive compounds intended to be used in food (Figure 3). In Europe, the EC Regulation No. 178/2002 and the Codex Alimentarius dictate that food additives, particularly those derived from food waste, must adhere to specific standards.

If these additives do not comply, they are governed by the Novel Food EC Regulation No. 258/97 (1997). This legislation necessitates thorough safety evaluations [99]. While these European regulations were initially tailored for food waste, they can be considered also for LAB by-products or derivatives. In the US, the utilization of food byproducts as ingredients is restrictive, and the FDA, under the guidance of the Federal Food, Drug, and Cosmetic Act (FFDCA) and parts of the Public Health Service Act (PHSA 42 US C), ensures their safety [99]. When evaluating the bioactive compounds produced by LAB for their application in food products, several safety considerations are essential [100]. Firstly, the potential toxicological properties of these compounds need to be evaluated. A compound, besides its antimicrobial properties, should not pose any health risk to consumers. Secondly, the allergenic potential must be determined. It is essential to ensure that these compounds do not induce allergic reactions in sensitive individuals, as such reactions could be detrimental to consumer health. Furthermore, the purity of these bioactive compounds is crucial. Impurities can not only diminish the compound efficacy but may also introduce unintended health risks. Lastly, it is essential to assess potential exposure dangers, particularly in the context of prolonged use. A compound deemed safe for short-term use might still pose significant risks when consumed regularly over a prolonged period. Collectively, these evaluations ensure that LAB-associated bioactive compounds meet the highest standards of consumer safety. As we further explore the various classes of these compounds, bacteriocins serve as a prime illustration of the thorough safety assessments they undergo.

Nisin is the sole bacteriocin to have received official approval as food preservative by the regulatory bodies [101]. Numerous studies have been conducted to ascertain the safety of bacteriocins, mainly focusing on aspects such as their stability in the gastrointestinal tract and potential side effects. Notably, most bacteriocins are susceptible to degradation by proteolytic enzymes in the stomach and small intestine [102]. For example, Gough et al. [103] confirmed the complete degradation of nisin during gastric and small intestine transit. Yet, it is worth noting that data on long-term effects of bacteriocin exposure through food consumption is scarce [101].

While many bacteriocins, including nisin, undergo rapid degradation by proteolytic enzymes in the stomach and small intestine, making them safe for consumption, innovative techniques are being explored to enhance their stability and efficacy. Gough et al. [104] illustrated this by demonstrating that encapsulation of nisin inside two specific starch-based matrices can significantly improve its resistance to degradation in the upper gastrointestinal tract. Nonetheless, as we progress in enhancing stability of bacteriocins, the focus on their safety assessments becomes even more pronounced. There are currently limited in vitro and in vivo data addressing the safety and toxicity of such stabilized bacteriocins. Therefore, it becomes crucial to rigorously examine their immunogenicity, as well as any potential in vitro and in vivo toxic effects.

Delving further into specific bacteriocins, enterocin AS-48 is an exemplary case. According to Baños et al. [105], this bacteriocin demonstrated no toxicity upon acute exposure and also showed no significant detrimental effects on BALB/c mice, even when administered at dosages of 50, 100, or 200 mg/kg over 90 days.

Fermented foods have been a dietary staple throughout human history, appreciated for their diverse flavors and potential health benefits. However, ensuring the safety of these products is paramount, especially considering the wide array of fermentation processes. Some fermented foods, unlike those made using well-defined starter cultures in controlled conditions, can undergo spontaneous fermentation or be manufactured with inadequate control measures. These uncontrolled processes may present health risks alongside their many advantages. Factors such as the use of low-quality ingredients and inadequate hygiene practices during manufacturing can compromise the safety of fermented foods, potentially leading to foodborne illnesses or outbreaks [106].

One crucial aspect of fermentation is the quality of water used in the process, as it should be free from microbial contamination. In regions where water quality is a concern,

the use of contaminated water from rivers, for instance, can elevate the risk of pathogens like *E. coli* finding their way into fermented foods. Since fermented products often lack heat treatments, it is imperative to ensure the personal hygiene of individu-als involved in collecting, processing, packaging, and distributing these foods [107].

Several studies shed light on the risks associated with consuming fermented dairy products. A study conducted in Kenya analyzed milk samples collected from various regions and discovered that Suusac consumption carried inherent risks for consumers. The research identified the presence of numerous pathogens, with *E. coli* found in all samples and Shigella ssp detected in 88.1% of them [108]. These findings may be at-tributed to poor production processes, storage conditions, or sales environments. The risk of contamination in fermented dairy products made from cow's milk could stem from inadequate hygiene practices in both the milking environment and fermentation tanks [106].

For example, an analysis of Roub, a traditional fermented dairy product produced in the rural areas of Sudan from cow's milk, revealed high levels of S. aureus and coli-form bacteria in the samples [109]. Similarly, a study of 60 random samples of various types of commercial cheeses in Egypt, including soft and hard cheeses, detected the presence of biogenic amines such as histamine, which can adversely affect human health [110]. The high levels of molds, yeasts, and biogenic amines in these samples in-dicated subopti-mal production and storage conditions. Researchers suggest that miti-gating the risk of biogenic amines may be achieved by adding Bacillus polymyxa D05-1 during the manu-facturing of hard cheese [110]. This innovative approach demon-strates the potential of microbial solutions to enhance the safety and quality of fer-mented dairy products, further highlighting the ongoing efforts to explore new strate-gies for food quality control and consumer well-being.

In the complex journey of using LAB and their associated bioactive compounds for dairy and food products, safety remains crucial. While significant efforts have been made to understand and evaluate LAB strains, ensuring their suitability for consumption goes beyond traditional practices. Both LAB and their produced compounds, including bacteriocins, require rigorous safety evaluations. Regulatory frameworks offer substantial guidelines, but the dynamic nature of microbial genetics and the complexity of their interactions with foods necessitate continual research and updating of these guidelines. As our knowledge deepens, it becomes imperative to not only leverage the benefits of these microbes and their compounds but also diligently ensure our food systems' safety. Therefore, we offer a comprehensive approach to safety assessment, which extends beyond regulatory frameworks. Our work underscores the nuanced insights obtained from recent studies about LAB strains in real-world applications. We emphasize the significance of addressing practical safety evaluations, including strain-specific traits, genetic stability, and unexpected genetic elements. This multifaceted perspective brings a fresh outlook to ensuring the safety of dairy products, elevating the uniqueness of this section.

## 6. Genetic Engineering in Dairy: Enhancing LAB Performance

Fermentation, an ancient practice, has experienced transformative evolutions through scientific advancements. Historically, the unpredictable nature of wild-type microorgan-isms dominated the fermentation world. However, the modern dairy industry demands consistency, safety, and enhanced product quality. Enter genetic engineering as a solu-tion that promises to redefine how LAB and their associated compounds can benefit the dairy sector.

In past practices, fermentation primarily relied on wild-type microorganisms, which acted as natural unpredictable culinary experts. Although essential to historical techniques, these microorganisms could behave inconsistently in controlled environments, leading to inconsistent antimicrobial effectiveness in the resulting fermented products. Modern techniques, however, increasingly consider genetically modified organisms (GMOs) for their predictability and enhanced benefits. Genetically engineered strains, fine-tuned through techniques based on the recombinant DNA technology, offer optimized production

of antimicrobial compounds and enhanced organoleptic properties, bridging the gap between traditional wisdom and modern requirements [111,112]. Innovations such as the CRISPR-Cas system further enhance the accuracy of these modifications, opening doors to exciting possibilities in dairy fermentation [113].

LAB strains are central to this scientific narrative. Genetic modifications on these strains have significantly impacted fermented dairy products texture, flavor, and safety [111,114].

Figure 4 illustrates a systematic approach to use GM LAB for dairy food production. Starting from raw milk as the foundational source, a step of microbial selection takes place, encompassing diverse groups, specifically LAB, fungi, and yeast. Once microbial strains have been meticulously selected, the process moves into precision fermentation. This step involves the strategic transfer of specific genes related to the desired traits, notably enzymes, from fungi and yeast to LAB strains with GRAS status. Such operation ensures that the enhanced LAB combines the robust capabilities of fungi and yeast with their inherent safe attributes. Following this, an indispensable phase of product evaluation must take place. This step evaluates the dairy product on multiple fronts, encompassing safety checks, bioactivity assessment, and other pertinent examinations, ensuring that the GMO-containing product maintains its efficacy and safety for consumption. Concluding the process, the potential product market should be assessed, emphasizing its feasibility and acceptability in consumer's perception. Figure 4 offers a holistic, step-by-step guidance for integrating GM LAB into the dairy sector, balancing innovation and safety.

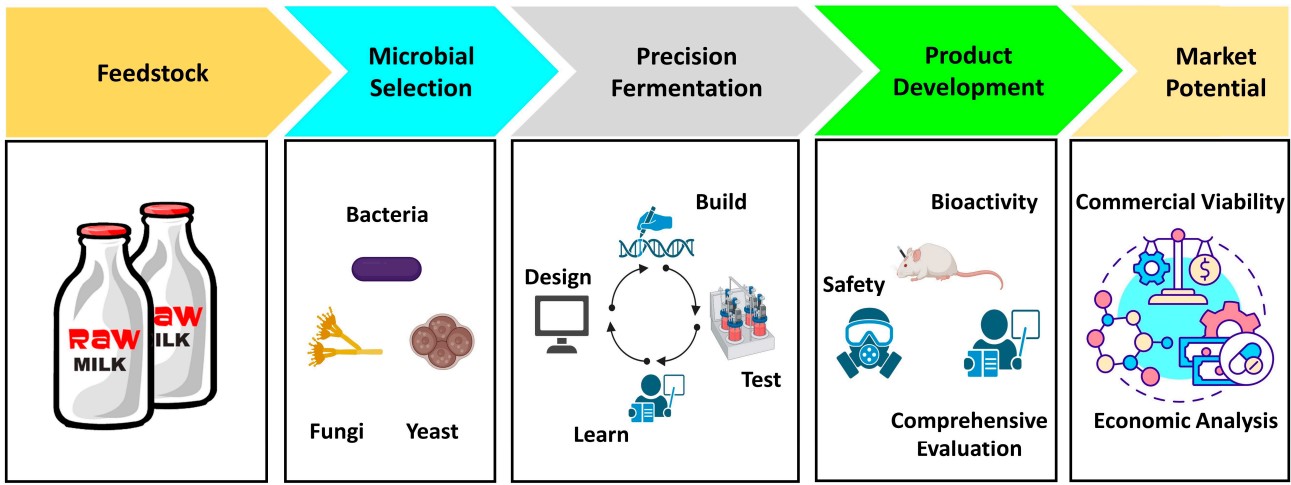

**Figure 4.** Systematic approach to integrating GM LAB in dairy production.

Beyond genetic enhancements, both nutritional and non-nutritional external factors play important roles. Saraniya and Jeevaratnam's exploration into optimal media components highlights how elements such as carbon and nitrogen sources crucially influence bacteriocin production. A deeper understanding of microbial nutritional requirements is as essential as genetic interventions to achieve desired fermentation outcomes [115,116].

As we use advanced capabilities in genetic modification, it is imperative to approach with a heightened sense of responsibility. The domain of GM LAB is intricate and has significant safety implications. Potential risks encompass the unintentional dissemination of GMOs into our ecosystems and the possible acceleration of antimicrobial resistance (AMR) bacteria diffusion. Such occurrences could accidentally shift the balance of microbial populations or confer the resistance advantages to pathogenic strains. If not properly evaluated, the effects of introducing GMOs into the environment can be unpredictable and vast. Ecosystems rely on a delicate balance of microbial interactions, and even slight interventions can lead to unforeseen consequences [91].

Moreover, as we navigate the challenge of rising AMR globally, the last thing we need is inadvertently bolstering these AMR strains through our scientific interventions. Thus, it becomes crucial to have a multi-pronged strategy in place. Biocontainment methods

provide a shield, ensuring that GMOs are confined to controlled environments and do not accidentally spread. Together, these strategies signify our commitment to innovation without compromising safety and ecological balance [91].

While the potential of GM LAB in the dairy sector is undeniable, ethical considerations surrounding their use are paramount. We must ponder not only the direct ecological impact but also the ethical implications of consumer rights, choice, and transparency. For instance, consumers have the right to be informed about the nature of their products, which necessitates clear labeling and communication strategies about GM LAB in dairy products. Furthermore, studying specific cases where GM LAB are being used would be beneficial. For instance, the modified organisms may interact with native species, leading to unforeseen ecological changes. Such examples serve as cautionary tales, emphasizing the importance of rigorous testing, monitoring, and containment [91,117].

Socio-economic implications are also worth exploring. GM LAB technologies could have various impacts on different stakeholders in the dairy sector—from producers to consumers. For instance, while some farmers might benefit from enhanced yields or product quality, others might be marginalized if they cannot access or afford such technologies. Lastly, evaluating the broader environmental footprint of GMO practices in the dairy sector is essential, especially when comparing them to traditional farming practices in aspects such as resource consumption, emissions, and waste generation [118].

In conclusion, the convergence of genetic engineering with dairy fermentation holds enormous potential. Furthermore, the emerging field of synthetic biology and metabolic engineering adds another layer of innovation to genetic engineering in the dairy industry. These cutting-edge approaches enable the precise design and construction of genetic circuits in LAB strains, allowing for tailored and intricate control of antimicrobial compound production. The ability to engineer LAB at a genetic level opens doors to the creation of strains that can synthesize specialized antimicrobial peptides with unique properties. This level of customization can result in dairy products with enhanced safety, novel flavors and textures that cater to evolving consumer preferences. By applying the full potential of genetic engineering in dairy fermentation, the industry can continue to adapt to changing demands while maintaining the highest standards of product quality. As we stand at the intersection of tradition and innovation, the future of fermented dairy products looks both flavorful and safe.

## 7. Conclusions

In the evolution of the dairy industry, the balance struck between time-honored traditions and progressive scientific advancements is evident. LAB stand at the forefront of this interplay, with their antimicrobial compounds production playing a pivotal role. These compounds not only ensure the preservation of dairy products but also actively promote human health. As we grapple with global challenges such as AMR, the significance of LAB's capabilities in producing these compounds has become more pronounced than ever. While traditional fermentation methods laid the foundation, contemporary advancements such as microencapsulation, nanotechnology, and predictive modeling have emerged to optimize and harness LAB's antimicrobial potential more effectively. These technologies promise enhanced effectiveness and precise delivery of antimicrobial agents. The realm of genetic engineering holds immense potential, promising heightened antimicrobial production. However, it is a double-edged sword, ushering in both opportunities and inherent challenges. It mandates the industry to tread carefully, always prioritizing safety and ecological equilibrium. A holistic understanding of various factors that influence LAB efficacy is imperative. To encapsulate, the dairy industry trajectory is a testament to the symbiotic relationship between age-old practices and cutting-edge science. It paints a promising picture of a future where dairy products not only tantalize the taste buds but are also fortified with potent antimicrobial defenses.

**Author Contributions:** Conceptualization, A.T.; writing—original draft preparation, A.T. and N.A.; investigation, N.A., A.T. and S.P.; writing—review and editing, A.T., A.G. and V.C.; supervision, A.G. All authors have read and agreed to the published version of the manuscript.

**Funding:** This research received no external funding.

**Institutional Review Board Statement:** Not applicable.

**Informed Consent Statement:** Not applicable.

**Data Availability Statement:** All data are presented in the paper.

**Conflicts of Interest:** The authors declare no conflict of interest.

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
