# Peer review of "Lactic Acid Bacteria in Dairy Foods: Prime Sources of Antimicrobial Compounds"

_fermentation, doi:10.3390/fermentation9110964_

Round 1

Reviewer 1 Report

Comments and Suggestions for Authors

fermentation-2642800

The topic developed in the review is interesting. However, there are many reviews on this topic, and although some are not so recent, I consider that they should be included in their references and highlight what they contribute and make a difference from this review.

Other comments that should be clarified:

Lines 34 - 35 - review, you don't understand what you want to highlight

Line 48 - the authors should use the term microbiota. This should be reviewed throughout the manuscript.

line 61 - review the author guide for citations in the text (the number is missing)

lines 63-65 - what the authors want to highlight is not fully understood. As bacteriocins are peptides, they will be affected by proteases,

line 108 - examples of LAB and what bacteriocins/classes they produce should be included.

lines 124-125 - which are approved/permitted to be used this way? Some details are mentioned in other sections later, but the information or the way it is presented should be organized

lines 179-180 - review. What do you mean by maintaining its activity during those conditions?

lines 192-193 - I consider that it is necessary to delve into the applications that have been successful or not against organisms of interest in various dairy products

line 203 - LAB can operate as either homo-fermenters or hetero-fermenters; this should be clarified in more depth

lines 243-244 - other acids? Such as PLA, etc.

lines 252-253 - provide more information about the evaluated products

line 256 - review the way to report citations in the text

line 271 - expand the information related to these processes

The review's contributions are unclear; the role that LAB plays in fermented dairy products needs to be clarified and deepened, not only as possible producers of antimicrobial agents.

Some references that should be reviewed to enrich the manuscript:

-          Reis, J. A., Paula, A. T., Casarotti, S. N., & Penna, A. L. B. (2012). Lactic acid bacteria antimicrobial compounds: characteristics and applications. Food Engineering Reviews, 4, 124-140.

-          Arqués, J. L., Rodríguez, E., Langa, S., Landete, J. M., & Medina, M. (2015). Antimicrobial activity of lactic acid bacteria in dairy products and gut: effect on pathogens. BioMed research international, 2015.

-          Ibrahim, S. A., Ayivi, R. D., Zimmerman, T., Siddiqui, S. A., Altemimi, A. B., Fidan, H., ... & Bakhshayesh, R. V. (2021). Lactic acid bacteria as antimicrobial agents: Food safety and microbial food spoilage prevention. Foods, 10(12), 3131.

Comments on the Quality of English Language

the quality of English used is adequate

Author Response

Dear Reviewer

We appreciate the positive feedback on our manuscript. We have reviewed the comments and addressed each of them throughout the document. We have also considered and cited the foundational references as requested by the reviewer. 

Reviewer 1

The topic developed in the review is interesting. However, there are many reviews on this topic, and although some are not so recent, I consider that they should be included in their references and highlight what they contribute and make a difference from this review.

Answer: We appreciate the positive feedback from the reviewer on our manuscript. We have reviewed the comments and addressed each of them throughout the document. We have also considered and cited the foundational references as requested by the reviewer.

Other comments that should be clarified:

Lines 34 - 35 - review, you don't understand what you want to highlight

Answer: We meant fermented foods often require less cooking or heating compared to their non-fermented counterparts, leading to energy savings. We have rephrased the sentence to clarify the point.

Line 48 - the authors should use the term microbiota. This should be reviewed throughout the manuscript.

Answer: Done, we have used the proper term throughout the manuscript.

line 61 - review the author guide for citations in the text (the number is missing)

Answer: Here, the content is referred to the citation 11 in line 60. We have gone through the citation in the manuscript and corrected the existing errors.

lines 63-65 - what the authors want to highlight is not fully understood. As bacteriocins are peptides, they will be affected by proteases,

Answer: We understand and agree that this is a known and expected phenomenon. Our intention was to emphasize the potential advantages of using bacteriocin-producing bacteria directly in dairy products to mitigate the challenges associated with purified bacteriocins sensitivity to proteolytic enzymes. To clarify this in our manuscript, we have rephrased the section in question to better convey our central idea. We believe that by employing bacteriocin-producing bacteria directly in dairy products, the continuous production of bacteriocins within the food matrix can counteract their degradation by naturally occurring proteases. The following content was added:

“For instance, in a study by Krishnamoorthi et al., nisin, a bacteriocin produced by Lactococcus lactis subsp. lactis strain CH3, was examined for its stability against various enzymes, including proteinase K and trypsin. As anticipated, given that bacteriocins are peptides, the bacteriocin lost its antibacterial activity upon treatment with these proteolytic enzymes [12]. The potential advantages of using bacteriocin-producing bacteria directly in dairy products could be an effective solution. Employing these bacteria can bypass the challenges associated with the inherent sensitivity of purified bacteriocins to proteolytic enzymes, as the continuous production of bacteriocins in the food matrix can counteract their degradation.”

line 108 - examples of LAB and what bacteriocins/classes they produce should be included.

Answer: Done, we have added more information in lines 121-134.

lines 124-125 - which are approved/permitted to be used this way? Some details are mentioned in other sections later, but the information or the way it is presented should be organized.

Answer: We acknowledge the need for clarity on this matter. To address this, we have revised the section in question to provide a clearer, more organized presentation of the strains and their respective impacts on flavor profiles. We believe this enhancement will offer readers a more comprehensive understanding without needing to cross-reference other parts of the manuscript. The following revision was added:

Moreover, the use of LAB in food products can greatly influence flavor profiles. Specific strains of LAB, such as Streptococcus thermophilus and L. lactis, are approved and commonly used to enhance flavors in dairy products. These strains are known to introduce a tangy and pleasant acidic taste, often associated with fermented dairy products like yogurt. On the other hand, certain strains, like Limosilactobacillus fermentum and Limosilactobacillus reuteri, if not used judiciously, can produce flavors that may be perceived as off or less palatable. This can be attributed to the production of certain metabolites such as biogenic amines (e.g., histamine), diacetyl (which in excess can impart a strong buttery flavor), or acetic acid (which can give a sharp, vinegar-like taste).

lines 179-180 - review. What do you mean by maintaining its activity during those conditions?

Answer: The phrase "retaining their efficacy" in this context refers to the bacteriocins' ability to maintain their antimicrobial or preservative function even after being exposed to high temperatures. We have revised and clarified it in the text.

lines 192-193 - I consider that it is necessary to delve into the applications that have been successful or not against organisms of interest in various dairy products.

Answer: Thank you for your suggestion. We have considered your suggestion and discussed more studies that applied different bacteriocins such as leucocin A and sakacin A in different dairy products such as Ricotta cheese & Mozzarella cheese. Please refer to lines 201 to 209.

line 203 - LAB can operate as either homo-fermenters or hetero-fermenters; this should be clarified in more depth.

Answer: Done, we have included more information on this matter in lines 245-299 as well as the relevant citations to clarify the concept:

lines 243-244 - other acids? Such as PLA, etc.

Answer: Thanks for the suggestion, we have added a paragraph and mentioned other important acids as well. The content has been added in lines 313-322.

lines 252-253 - provide more information about the evaluated products

Answer: Done, we have added more information about the products in the section requested.

line 256 - review the way to report citations in the text

Answer: Done, we have gone through the citations in the text and corrected the errors.

line 271 - expand the information related to these processes

Answer: Done, we have expanded on fed-batch and simultaneous saccharification and fermentation as requested.

The review's contributions are unclear; the role that LAB plays in fermented dairy products needs to be clarified and deepened, not only as possible producers of antimicrobial agents.

Answer: We appreciate the feedback provided. Our original intent for this review was to primarily spotlight the production of antimicrobial compounds by LAB in dairy foods, emphasizing both the challenges and solutions inherent to this topic. Recognizing the reviewer's concern about delineating the broader role of LAB in fermented dairy products, we have opted to expand the scope of the manuscript in the revised version in places that the reviewer requested more discussion. While our primary focus on antimicrobial activity remains, we have now incorporated a more comprehensive overview of LAB's multifaceted functions and contributions to fermented dairy products.

Some references that should be reviewed to enrich the manuscript:

-          Reis, J. A., Paula, A. T., Casarotti, S. N., & Penna, A. L. B. (2012). Lactic acid bacteria antimicrobial compounds: characteristics and applications. Food Engineering Reviews, 4, 124-140.

-          Arqués, J. L., Rodríguez, E., Langa, S., Landete, J. M., & Medina, M. (2015). Antimicrobial activity of lactic acid bacteria in dairy products and gut: effect on pathogens. BioMed research international, 2015.

-          Ibrahim, S. A., Ayivi, R. D., Zimmerman, T., Siddiqui, S. A., Altemimi, A. B., Fidan, H., ... & Bakhshayesh, R. V. (2021). Lactic acid bacteria as antimicrobial agents: Food safety and microbial food spoilage prevention. Foods, 10(12), 3131.

Answer: Thank you for your suggestion. We have considered and discussed these references in the revised manuscript in the section of “antimicrobial compounds diversity in dairy foods.”

Best regards

Reviewer 2 Report

Comments and Suggestions for Authors

This review deeply describes and assesses the antimicrobial properties of lactic acid bacteria, and their antimicrobial compounds, finishing with genetic engineering enhancing these properties. The manuscript is greatly organized and comprehensively written in high-quality English. I suggest avoiding using "et al" in the list of references. Instead, I suggest to list all authors. 

Author Response

Dear Reviewer

We appreciate the positive feedback on our manuscript. We have

reviewed the journal format for citation and revised it accordingly.

Best regards

Reviewer 3 Report

Comments and Suggestions for Authors

Abstract

1.       Consider providing specific examples or highlights within each section to make the abstract more engaging. For instance, mention a few key antimicrobial compounds or the challenges related to stability.

Introduction

2.       Overall, the introduction sets a strong foundation for the review by introducing the key concepts and objectives. However, the addition of citations and specific references to previous research would further enhance its credibility.

Body

3.       Overall, the document provides some but not well organized-structured and informative overview of bacteriocins and their relevance in dairy food preservation.

4.       It effectively presents classification, strategies, practical examples, and specific details about nisin, making it a valuable resource for those interested in this topic.  need to cite all important references...

5.       It also covers both the scientific aspects and practical applications of organic acids in the food industry, making it valuable for professionals and researchers in food science and technology. however, the writing lack of organization and information was present in a very random fashion.

6.ta There is a need to provide a more comprehensive overview of the safety considerations surrounding LAB and bioactive compounds in dairy foods.

7.       It effectively combines theoretical information with practical case studies to underscore the importance of safety assessments is presented but missing important points...

8.       The document provides a valuable overview of the potential applications of genetic engineering in the dairy industry.

9.       However, it could benefit from improved citation practices, more concrete examples, and a deeper exploration of ethical and environmental considerations. Additionally, some minor language and style improvements would enhance overall readability.

10.   Additionally, it's essential to acknowledge that the safety of LAB strains and bioactive compounds can vary, and ongoing research is necessary to address emerging concerns.

Conclusion

The conclusion was not well structured and missing important points.

additional comments are attached.

Comments on the Quality of English Language

ok

Author Response

Dear Reviewer

We appreciate the feedback on our manuscript. We have reviewed the comments and addressed each of them throughout the document. Moreover, the manuscript language has been evaluated by a native English speaker.

Abstract

  1. Consider providing specific examples or highlights within each section to make the abstract more engaging. For instance, mention a few key antimicrobial compounds or the challenges related to stability.

Answer: We appreciate the feedback from the reviewer on our manuscript. We have reviewed the comments and addressed each of them throughout the document. We have also considered the comment on the abstract and within the manuscript included the information the reviewer requested. Moreover, the manuscript language has been evaluated by a native English speaker.

  1. Overall, the introduction sets a strong foundation for the review by introducing the key concepts and objectives. However, the addition of citations and specific references to previous research would further enhance its credibility.

Answer: Done, we have added more references to the manuscripts as requested by the reviewers and enriched the discussion part in different sections.

Body

  1. Overall, the document provides some but not well organized-structured and informative overview of bacteriocins and their relevance in dairy food preservation.

Answer: We have gone through the section and added more organized information by adding more examples of bacteriocins and LAB in several dairy products and discussing their effectiveness. Please see the lines 124-137 & 205-212.

  1. It effectively presents classification, strategies, practical examples, and specific details about nisin, making it a valuable resource for those interested in this topic.  need to cite all important references...

Answer: Thank you for the positive feedback. We've reviewed the section and enriched it with more structured details, incorporating additional examples of bacteriocins and LAB found in various dairy products and delved into their efficacy as well as the references.

  1. It also covers both the scientific aspects and practical applications of organic acids in the food industry, making it valuable for professionals and researchers in food science and technology. however, the writing lack of organization and information was present in a very random fashion.

Answer: Thank you for your guidance. Based on your feedback, we have updated the section, providing detailed insights on LAB and their metabolites, focusing on both homo and hetero fermentation. We've expanded our discussion to highlight the diversity of organic acids produced by LAB and their respective antimicrobial mechanisms. For instance, beyond acetic acid, Phenyllactic Acid (PLA) is synthesized from phenylalanine and exhibits broad antifungal properties, affecting fungal cell membranes and energy metabolism. We've also addressed other organic acids like lactic acid, formic acid, and propionic acid, elucidating their unique contributions to antifungal activity. Particularly, formic and propionic acids infiltrate fungal cells, altering their internal pH and metabolic processes. Understanding this wide-ranging antifungal toolkit of LAB, which includes acetic acid, PLA, and other organic acids, is pivotal. Their combined and potentially synergistic impacts can notably improve the preservation and safety of dairy products. We've also ensured the incorporation of the necessary references.

  1. There is a need to provide a more comprehensive overview of the safety considerations surrounding LAB and bioactive compounds in dairy foods.

Answer: In our manuscript, we have dedicated sections discussing international guidelines related to LAB and their bioactive compounds in dairy products. Furthermore, we have delved deep into the current literature to elucidate the ongoing studies and their implications on safety considerations. While we believe we have provided a comprehensive overview, we appreciate your expertise and insights. We would appreciate if the reviewer specifies any particular areas or points within the topic of safety considerations that the reviewer believes need to be added. This would greatly assist us in refining and augmenting the content to ensure that our manuscript addresses all pertinent concerns.

  1. It effectively combines theoretical information with practical case studies to underscore the importance of safety assessments is presented but missing important points.

Answer: Thank you for drawing attention to the balance of theoretical information and practical case studies in our manuscript, especially concerning the importance of safety assessments. We have endeavored to integrate both to provide a well-rounded understanding of the subject. We take your feedback seriously and are committed to ensuring our manuscript is both comprehensive and clear. However, it would be immensely helpful if you could provide specific insights or pointers on which "important points" you feel are missing from our discussion. This would allow us to address and incorporate them effectively.

  1. The document provides a valuable overview of the potential applications of genetic engineering in the dairy industry.

Answer: We appreciate the positive feedback from the reviewer!

  1. However, it could benefit from improved citation practices, more concrete examples, and a deeper exploration of ethical and environmental considerations. Additionally, some minor language and style improvements would enhance overall readability.

Answer: Thank you for your valuable feedback regarding the need for a deeper exploration of the environmental considerations of GMO practices in the dairy sector. In response, we have enriched our discussion by evaluating the broader environmental footprint of GMO practices. Specifically, we have drawn comparisons with traditional farming practices, focusing on aspects such as resource consumption, emissions, and waste generation. We believe this addition provides a more comprehensive view of the topic, addressing the concerns you highlighted. Please refer to lines 692-707. The manuscript has also been evaluated by a native speaker.

.Additionally, it's essential to acknowledge that the safety of LAB strains and bioactive compounds can vary, and ongoing research is necessary to address emerging concerns.

Answer: Thank you for the suggestion. The point is stressed and addressed in lines 624-632.

Conclusion

The conclusion was not well structured and missing important points.

Answer: In response to your suggestions, we have revised the conclusion to be more structured and encompassing. We've aimed to highlight the pivotal role of Lactic Acid Bacteria (LAB) in the dairy industry, from traditional practices to modern scientific advancements. The updated conclusion also delves deeper into the challenges and opportunities presented by technological advancements and the nuances of genetic engineering. We believe this revised conclusion captures the essence of our research and provides a cohesive summary of our findings.

Best regards

Round 2

Reviewer 1 Report

Comments and Suggestions for Authors

The authors took into account the comments and suggestions to improve their manuscript. This version is better understood and more comprehensible to read.

Comments on the Quality of English Language

none

Author Response

Thank you!

Reviewer 3 Report

Comments and Suggestions for Authors

the manuscript was rejected earlier....

Author Response

NA